# Human genome diversity data reveal that L564P is the predominant TPC2 variant and a prerequisite for the blond hair associated M484L gain-of-function effect

Julia Böck[1][◉], Einar Krogsaeter[1][◉], Marcel Passon[1], Yu-Kai Chao[1], Sapna Sharma[2], Harald Grallert[2], Annette Peters[2], Christian Grimm[1]*

1 Walther Straub Institute of Pharmacology and Toxicology, Faculty of Medicine, Ludwig-Maximilians-Universität, Munich, Germany, 2 Helmholtz Zentrum–Deutsches Forschungszentrum für Gesundheit und Umwelt (GmbH), Institute of Epidemiology, Neuherberg, Germany

◉ These authors contributed equally to this work.

* christian.grimm@med.uni-muenchen.de

**Data Availability Statement:** All relevant data are within the manuscript and its Supporting Information files.

## Abstract

The endo-lysosomal two-pore channel (TPC2) has been established as an intracellular cation channel of significant physiological and pathophysiological relevance in recent years. For example, TPC2$^{-/-}$ mice show defects in cholesterol degradation, leading to hypercholesterinemia; TPC2 absence also results in mature-onset obesity, and a role in glucagon secretion and diabetes has been proposed. Infections with bacterial toxins or viruses e.g., cholera toxin or Ebola virus result in reduced infectivity rates in the absence of TPC2 or after pharmacological blockage, and TPC2$^{-/-}$ cancer cells lose their ability to migrate and metastasize efficiently. Finally, melanin production is affected by changes in hTPC2 activity, resulting in pigmentation defects and hair color variation. Here, we analyzed several publicly available genome variation data sets and identified multiple variations in the TPC2 protein in distinct human populations. Surprisingly, one variation, L564P, was found to be the predominant TPC2 isoform on a global scale. By applying endo-lysosomal patch-clamp electrophysiology, we found that L564P is a prerequisite for the previously described M484L gain-of-function effect that is associated with blond hair. Additionally, other gain-of-function variants with distinct geographical and ethnic distribution were discovered and functionally characterized. A meta-analysis of genome-wide association studies was performed, finding the polymorphisms to be associated with both distinct and overlapping traits. In sum, we present the first systematic analysis of variations in TPC2. We functionally characterized the most common variations and assessed their association with various disease traits. With TPC2 emerging as a novel drug target for the treatment of various diseases, this study provides valuable insights into ethnic and geographical distribution of TPC2 polymorphisms and their effects on channel activity.

**Funding:** This work was supported, in part, by funding of the German Research Foundation DFG (SFB/TRR152 project P04 to CG and GR-4315/4-1 to CG) and by the NCL (Neuronal Ceroid Lipofuscinosis) Foundation, Hamburg, Germany. The funders had no role in study design, data collection and analysis, decision to publish, or preparation of the manuscript.

**Competing interests:** The authors declare that they have no conflict of interest.

## Author summary

The endo-lysosomal cation channel TPC2 is implicated in numerous human diseases ranging from metabolic disease, Parkinson's disease, cancer and pigmentation defects, to infectious diseases such as Ebola, Covid-19, and bacterial infections. Here, we present a functional analysis of several polymorphisms occurring in the human TPC2 protein in distinct populations. By evaluating several human genome databases, we identified a large number of single nucleotide polymorphisms in TPC2. We electrophysiologically characterized the most common polymorphisms by applying the endo-lysosomal patch-clamp technique. We thereby identified several novel gain-of-function variants and found the TPC2 variation L564P to be a prerequisite for the previously described M484L gain-of-function effect associated with blond hair. In addition, publicly available genome-wide association study databases were assessed, and linked traits of the investigated TPC2 polymorphisms interrogated. Considering that different human populations have a different likelihood of carrying the identified gain-of-function variations, these findings appear highly relevant for further assessment of TPC2 as a pharmacological drug target.

## Introduction

DNA variants are traditionally divided into DNA mutations and single nucleotide polymorphisms (SNPs), based on their occurrence in <1% and >1% of the population, respectively. Variations with a proven effect on gene function are also termed functional polymorphisms. Such functional polymorphisms may impact development of disease or response to pathogens, chemicals, vaccines or drugs. This is particularly important in the context of personalized medicine, as gene variation can affect, e.g. drug metabolism or side effects. Biomedical research is increasingly focusing on a better understanding of the functional relevance of such variations and their association with diseases and therapies. For instance, variations associated with a certain disease in one population may appear harmless or are irrelevant in other populations. One famous example is malaria resistance in people carrying certain variations in hemoglobin, which on the one hand results in sickle-cell anemia, but on the other hand is a selection advantage, at least for heterozygous carriers, in geographic regions where malaria is prevalent [1,2]. As with other genetic variations, functional variations in TPC2 may also impact health and survival under certain environmental conditions and in certain geographical areas, while in others they might not.

The two-pore cation channel (TPC2) has recently emerged as an intracellular ion channel of significant physiological and pathophysiological relevance. TPC2 has been associated with various essential functions in endo-lysosomes, such as trafficking, autophagy, exocytosis, and lysosomal cation/pH homeostasis [3–6]. Accordingly, changes to its function bear consequences for numerous diseases, e.g. infectious diseases in which viruses or bacterial toxins are trafficked through the endo-lysosomal system [7–9], metabolic diseases caused by defects in endo-lysosomal trafficking or cargo degradation[10,11], cancer [12–16], or pathologies involving lysosome-related organelles such as platelet dense granules [17], melanosomes [18,19], or cytolytic granules [20]. Recently, two variations in TPC2 associated with hair pigmentation defects were found to be gain-of-function (GOF) variants: rs35264875 (encoding M484L) results in an increased sensitivity to the endogenous TPC2 ligand PI(3,5)P$_2$, while rs3829241 (encoding G734E) results in reduced channel inhibition by ATP [21]. Given their direct functional relevance to channel activity, and the link between channel function and various diseases, we predict TPC2 polymorphisms confer traits which can be acted upon by various

selection pressures. For example, TPC2 GOF polymorphisms could represent risk factors, enhancing endosomal trafficking and predisposing carriers to infections [7–9,22]. Further- more, since TPC2 activity has been implicated in glucose homeostasis, altered TPC2 activity may correlate with the development of diabetes mellitus [23–25].

In an attempt to assess the global distribution of TPC2 variations in humans, we analyzed several human genome datasets: the Simons Genome Diversity Project (SGDP) dataset, com- prising genomes of 279 individuals from 142 indigenous populations in Africa, Europe, Asia, Australia, Oceania, and America [26], the 1000 Genome Project dataset (1000GP, 2504 genomes) [27], and the gnomAD data set (~140.000 genomes) [28]. We screened these datasets for polymorphisms in TPC2 and other endo-lysosomal cation channels, namely TPC1, TRPML1, TRPML2, and TRPML3. We found that TPC2 variations are more frequent than variations in the other channels. One variation in TPC2, L564P, occurs much more frequently on a global scale than the nominal "wild-type" TPC2 isoform. Other variations were found to occur in more geographically restricted manners. Unexpectedly, we found L564P to be a pre- requisite for the blond hair-associated M484L variant to exert its GOF effect, as demonstrated by endo-lysosomal patch-clamp electrophysiology. Furthermore, additional variants were found to occur frequently in a homozygous manner with distinct ethnic or geographical distri- bution. While some of these variations had no apparent effect on channel activity, others showed increased sensitivity to the endogenous TPC2 ligand PI(3,5)$P_2$, akin to the previously characterized M484L polymorphism. These results reveal a functional diversity of TPC2 between distinct populations, with potential relevance in manifestation and progression of TPC2-associated diseases.

## Results

### L564P is the predominant TPC2 variant on a global scale

We used the data provided by SGDP, 1000GP, and gnomAD to assess the occurrence of varia- tions in the sequence of TPC2 and its related endo-lysosomal cation channels TPC1, TRPML1, TRPML2, and TRPML3 across different populations and ethnicities. In the SGDP dataset, 22 variations were identified for TPC2, while only 7 were found for TPC1 and TRPML1 each, 10 for TRPML2, and 5 for TRPML3 (Figs 1A and 1B and S1). One of the identified variations in TPC2, L564P, was found with particularly high frequency (Figs 1A, 1C, 1D, 2E and 2G). Other homozygous variations found to occur with increased frequency in TPC2 in the SGDP data set were V219I, K376R, G387D, M484L, and G734E. The polymorphisms V219I, M484L and G734E are frequently present in European populations (Fig 2A, 2D, 2F and 2G). G387D showed a low frequency across most populations (Fig 2C and 2G), while K376R (Fig 2B and 2G) showed comparably high frequencies across all populations with the exception of Native Americans, who over all showed low TPC2 variation rates in their genomes (Fig 2G). L564P occurs at a very high frequency in a homozygous manner on a global scale except for Native Americans and some African and Asian populations. Nearly all European SGDP samples were found to be homozygous for L564P, with only one sampled Hungarian being heterozygous for L564P. Similarly, in most Asian, African, Australian and Oceanian populations, the homozy- gous L564P variant predominates (Fig 2G).

Analysis of the 1000GP and the gnomAD datasets confirmed the SGDP dataset findings, including the very high frequency of the L564P variation across all populations (Figs 3 and S2 and S3). Like the SGDP dataset, the 1000GP dataset revealed a higher variation frequency in TPC2 (63 SNPs) compared to TPC1 (33 SNPs), TRPML1 (34 SNPs), TRPML2 (34 SNPs), and TRPML3 (31 SNPs) (S2 Fig). Accordingly, the gnomAD dataset supported these observations, revealing higher variation frequencies for TPC2 (412 SNPs) compared to TPC1 (381),

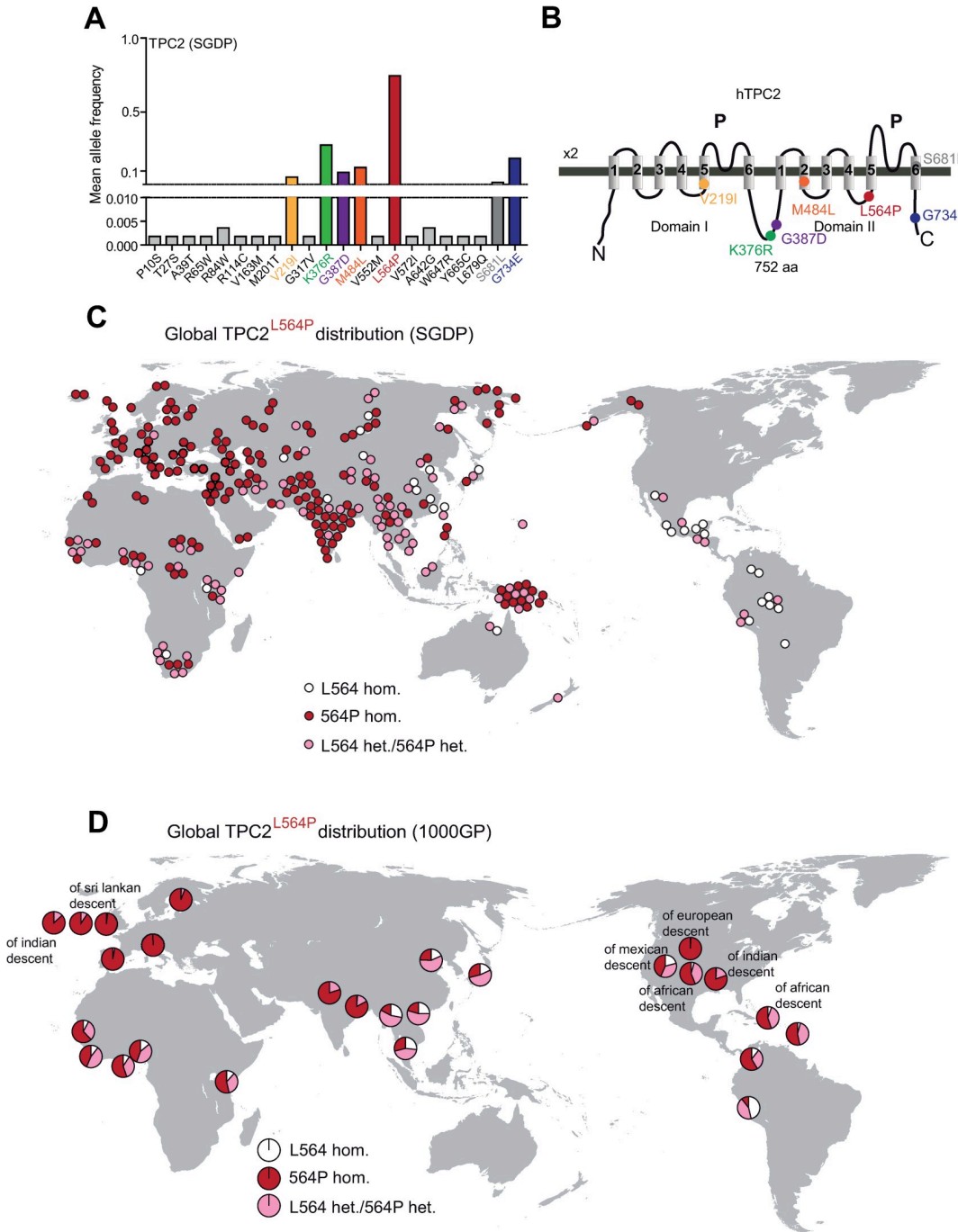

**Fig 1. Analysis of the sequencing data derived from the SGDP data set and the 1000GP data set.** (A) Mean allele frequency in the Simons Genome Diversity Project (SGDP) data set of all homozygous (colored) and heterozygous SNPs in the TPC2 gene leading to a missense mutation. (B) Cartoon showing the estimated positions of the polymorphisms in the human TPC2 protein. (C) Map of the worldwide distribution of TPC2[L564P] in the SGDP data set. Each circle represents one individual from an indigenous population, respectively. White circles represent individuals homozygous for TPC2[P564L], red circles represent individuals homozygous for TPC2[L564P], and pink circles represent heterozygous individuals. (D) Map of the worldwide distribution of TPC2[L564P] in the 1000GP data set. Each circle represents one of the 26 populations (indigenous or non-indigenous (other descent) as indicated. Color coding scheme corresponds to the one in (C).

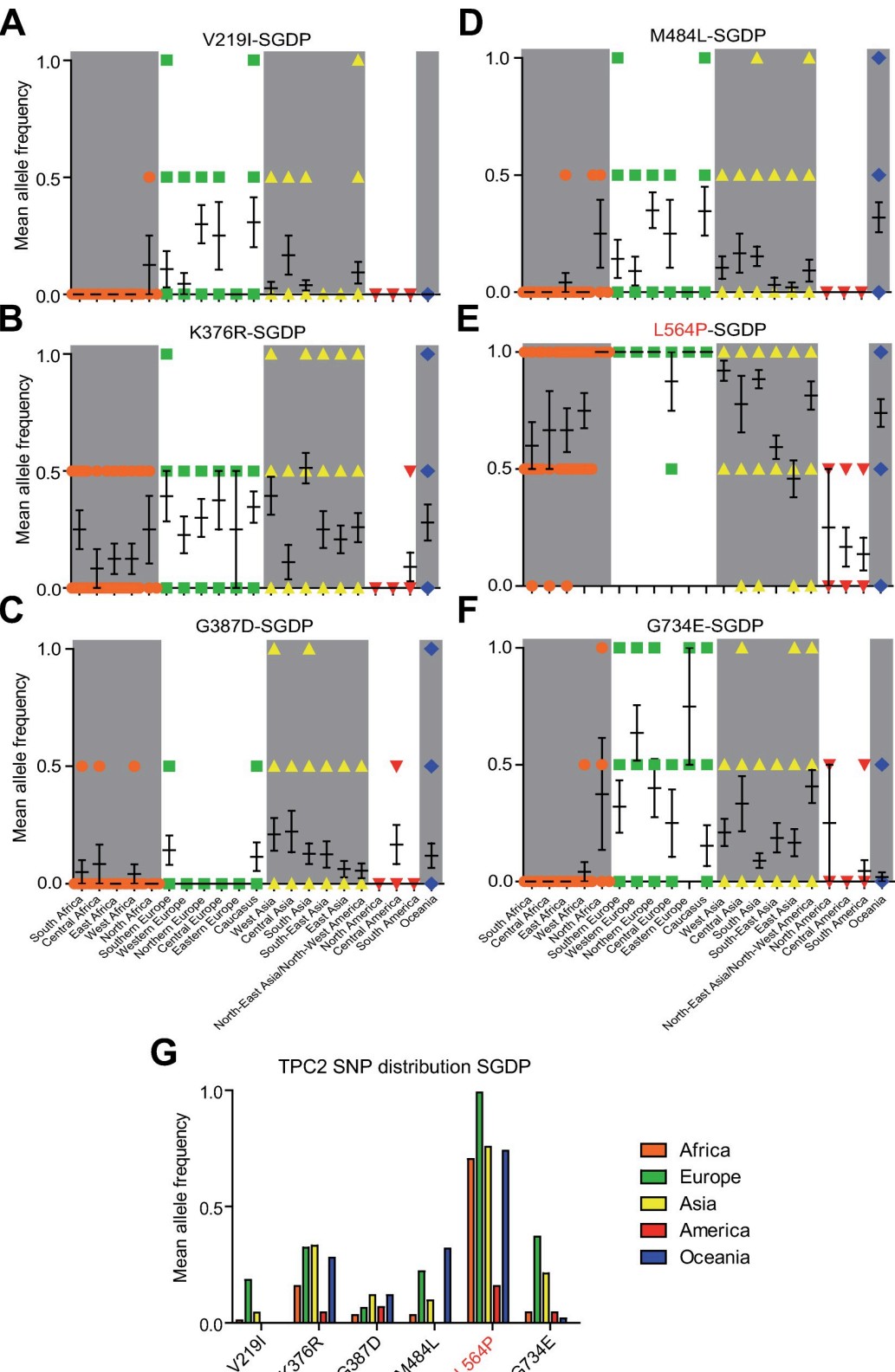

**Fig 2. Geographical distribution of all homozygous TPC2 SNPs found in the SGDP data set.** The 297 samples were grouped according to their geographic origin. (A-F) Mean allele frequency of each SNP (mean ± SEM) grouped by geographic region, each symbol (circle, square, triangle) represents one sample (G) Mean allele frequency of each SNP grouped by continent.

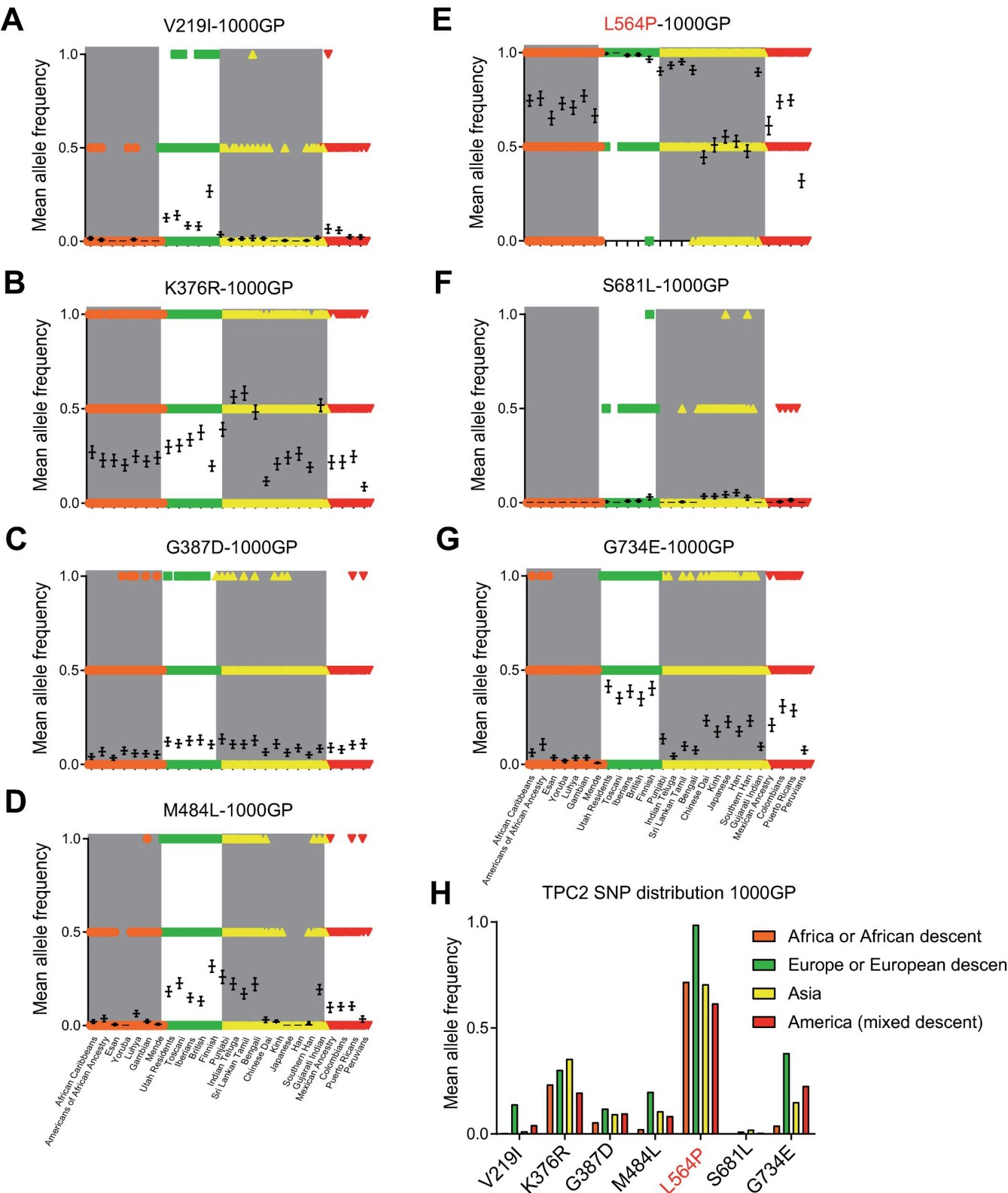

**Fig 3. Geographical distribution of all homozygous TPC2 SNPs found in the 1000GP data set.** The 2054 samples were grouped according to geographic origin. (A-G) Mean allele frequency of each SNP (mean ± SEM) in each population grouped by geographic region, each symbol (circle, square, triangle, rhombus) represents one sample. (H) Mean allele frequency of each SNP grouped by continent.

TRPML1 (272), TRPML2 (278), and TRPML3 (260) (S3 Fig). In general, variations were more frequent in TPC2 compared to the other endo-lysosomal cation channels (S1–S3 Figs). In addition to the homozygously occurring variations in TPC2 in the SGDP dataset (V219I, K376R, G387D, M484L, L564P, and G734E), another variation, S681L, was found to occur in a homozygous manner in the 1000GP and gnomAD datasets. This variation occurs mainly in Han Chinese and Japanese samples (Figs 3 and S2 and S3). Homozygously occurring variations were less frequent in other endolysosomal ion channels. In the 1000GP dataset, only one homozygously occurring variation was found in TPC1 (G803W) and three in TRPML2 (V20I, M365V, and K370Q), but none in TRPML1 and TRPML3 (S2 Fig). Only in the gnomAD dataset, homozygously occurring variations were found for TRPML1 and TRPML3 (S3 Fig). Neither of the homozygously occurring TRPML1 variations are known to cause mucolipidosis type IV, a rare neurodegenerative lysosomal storage disease caused by certain TRPML1 mutations [29,30]. For TRPML2 and TRPML3, homozygous premature stop codon variants were detected, revealing the existence of people with complete loss of either TRPML2 or TRPML3 channels.

## TPC2 in ancient human genomes and in evolution

Next, we analyzed published genome sequencing data from ancient samples. In Neanderthal, Denisovan, and ancient modern humans, both L564 and 564P were present. Again, 564P occurred with higher frequency, suggesting that at least in the last 40.000 to 50.000 years, the occurrence and geographical distribution of L564 versus 564P was not very different from contemporary samples. An analysis of TPC2 variations in non-human primates revealed that all assessed primate reference genomes have the L564P variation, suggesting an evolutionary conserved high prevalence of 564P (S4 Fig).

## TPC2 variations have predominantly developed on the background of L564P

The structure of the human TPC2 channel was recently described, pinpointing amino acid 564 to be present on the five-residue 2TMD4/2TMD5 (IIS4-S5) loop, which is of importance in channel gating [31]. As position 564 is located on a critical part of the protein, we assessed channel function of the $TPC2^{L564P}$ in comparison to the "wildtype" variant, $TPC2^{P564L}$ by using the endo-lysosomal patch-clamp technique. Stimulation of both variants with the endogenous TPC2 agonist, $PI(3,5)P_2$, revealed no significant difference in channel conductance (Figs 4A, 4B and 5H). However, the genomic data analysis demonstrated that individuals homozygous for 564P have an increased likelihood of carrying additional homozygous variations in TPC2 (Fig 4C–4H). Only one SNP, S681L, predominantly found in East Asian populations, appeared associated with the L564 background (Fig 4H). Taking this into consideration, we re-evaluated the human donor fibroblast samples used by Chao et al. (2017) for studying the GOF variations of TPC2. We found that all fibroblasts were homozygous for the L564P polymorphism, and that the cDNA originally isolated from HEK293 cells used by Chao et al. (2017) was likewise homozygous for L564P [21]. Therefore, the question arose whether amino acid 564 may affect the recently described GOF variation of TPC2, namely $TPC2^{M484L}$.

## L564P is essential for the blond hair associated GOF effect of M484L

Surprisingly, in endo-lysosomal patch-clamp experiments we found that M484L only acts as a GOF variant on the 564P, but not on the L564, background (Fig 5A, 5B and 5H). Thus, $PI(3,5)P_2$-elicited currents in the $TPC2^{M484L/P564L}$ variant were not significantly different from $TPC2^{P564L}$ ("wild type") or $TPC2^{L564P}$, while $TPC2^{M484L/L564P}$ showed a GOF effect as reported

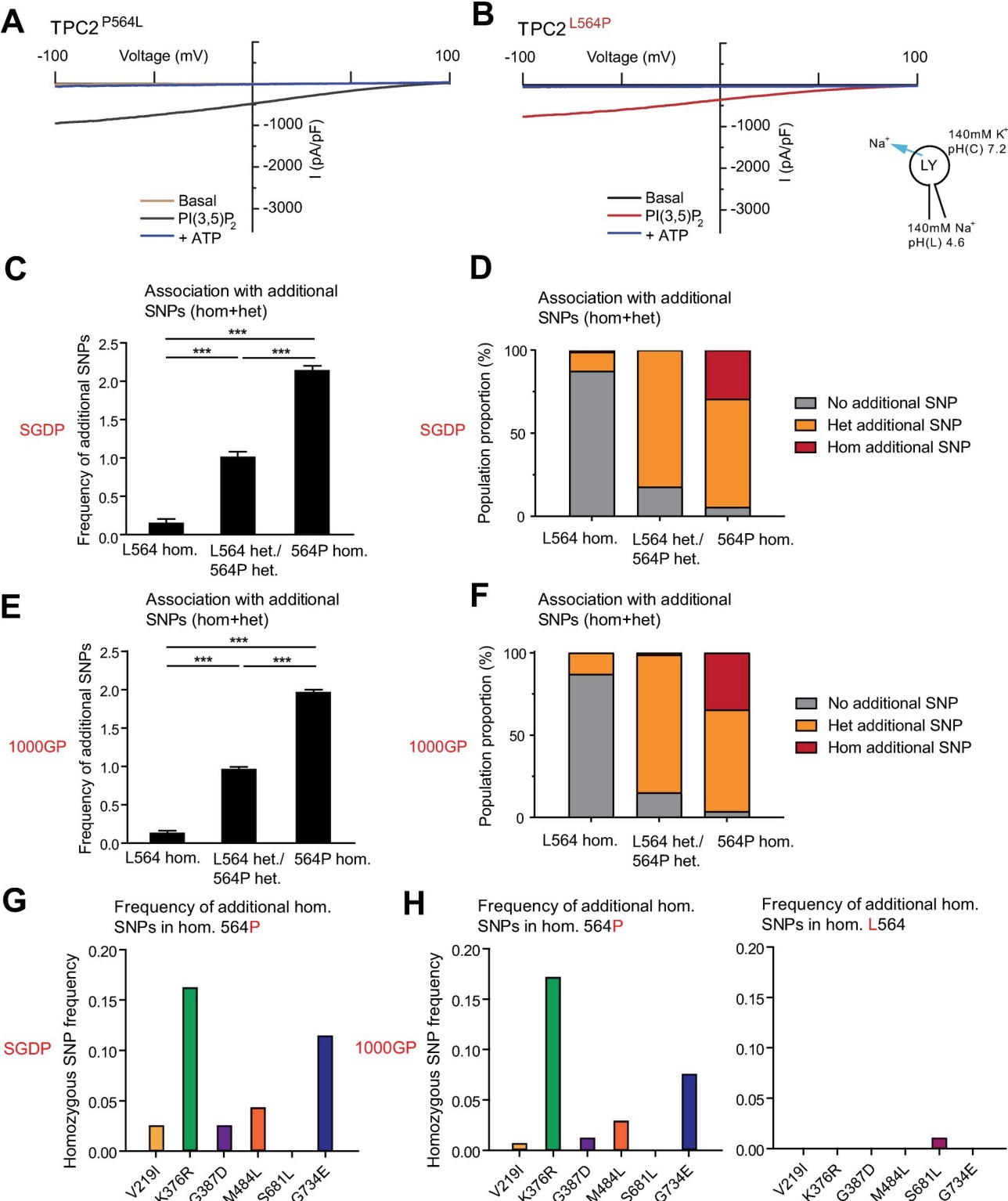

**Fig 4. TPC2$^{L564P}$ in association with other TPC2 SNPs.** (A, B) Representative PI(3,5)P$_2$ (1 μM) activated current densities in vacuolin-enlarged lysosomal vesicles expressing TPC2$^{L564P}$ or TPC2$^{P564L}$ and the corresponding currents after block with ATP (1 mM). (C-F) Frequency of other homozygous and heterozygous TPC2 SNPs on either L564 or 564P background. (C, D) Results from the SGDP data set. (E, F) Results from the 1000GP data set. (G-H) Frequency of homozygous SNPs on either L564 or 564P background in the SGDP data set or the 1000GP data set.

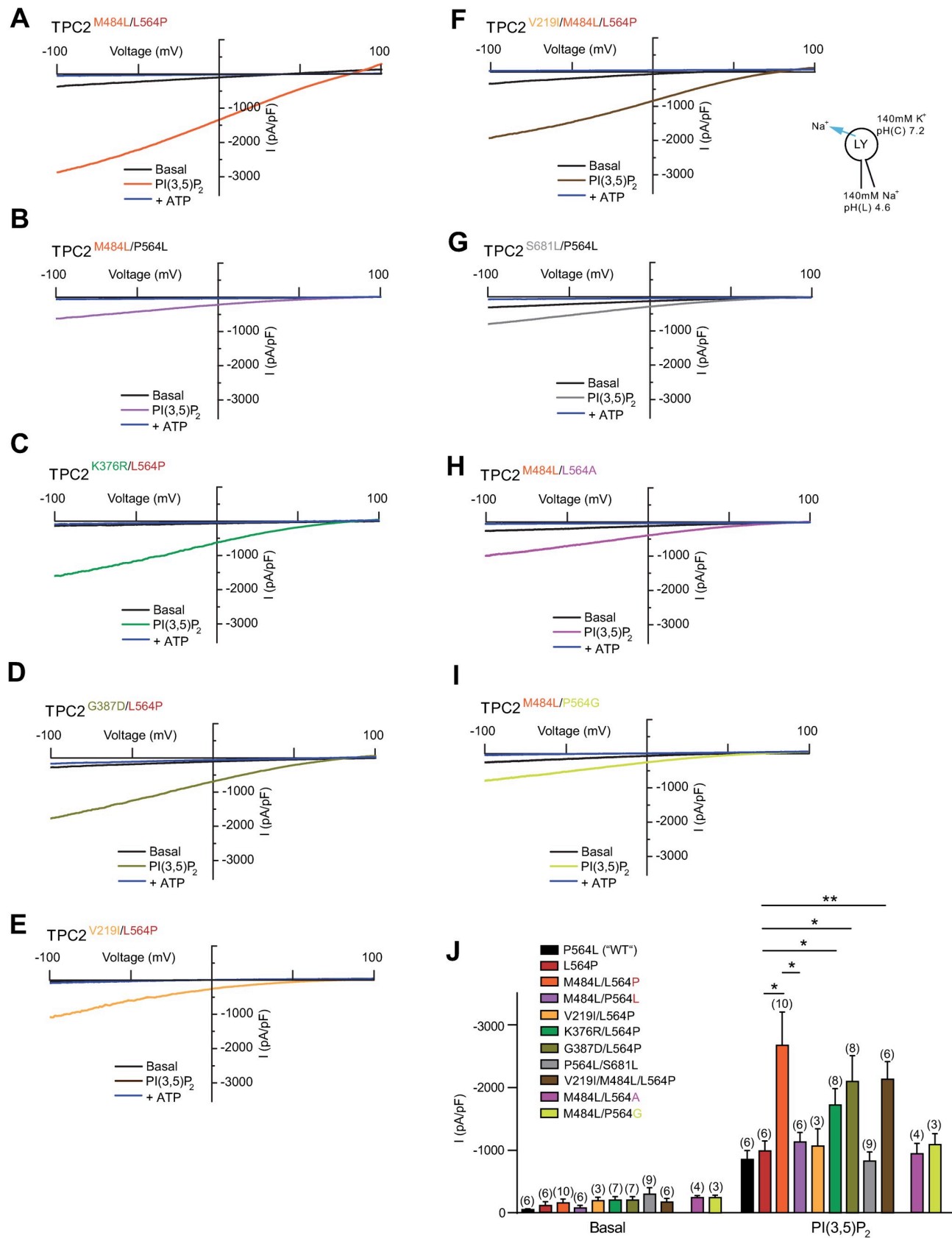

**Fig 5. Effect of PI(3,5)P$_2$ on different variations of TPC2.** (A-I) Representative PI(3,5)P$_2$ (1 μM) activated current densities in vacuolin-enlarged lysosomal vesicles isolated from HEK293 cells overexpressing different human TPC2 variants and the corresponding currents after block with ATP (1 mM). (J) Statistical summary of data as shown in A-I and in Fig 4A and 4B. Shown are average current densities (mean ± SEM) at -100 mV. Unpaired t-test were applied. *p < 0.05 and **p < 0.01.

previously [21]. To further elucidate the surprising finding that M484L only results in a GOF in combination with proline at position 564, we applied site-directed mutagenesis and found that introduction of prolines up- or downstream of 564, i.e., at position 563 or 565, did not result in GOF in combination with M484L (S7A and S7B Fig). Furthermore, we tested 564A and 564G in combination with M484L. As expected, alanine showed no GOF effect in combination with M484L. Also glycine, which may act as a helix breaker, albeit milder than proline, showed no GOF effect (Fig 5H–5J). These data suggest that only a strong helix breaking amino acid (P) but not A, L or G is capable of mediating the observed GOF effect in combination with 484L. Based on the recently resolved hTPC2 (M484/564P) cryo-EM structure [31] it is evident that L564P is situated directly between M484L and the channel pore. We therefore assume the M484L-associated GOF to be transduced through the L564P-encoding IIS4-S5 linker, dilating the channel pore further upon PI(3,5)P$_2$ activation. The IIS4-S5 linker has already been implicated in channel gating, as an extension of the IIS4-S5 linker appears necessary to provide space for pore dilation and channel opening. Substituting the helix-initiating 564P with a leucine would dramatically affect this linker helix extension. We therefore propose a model where M484L is amplifying the effect of PI(3,5)P$_2$ activation, requiring signal transduction through L564P to result in pore dilation. However, to confirm this, comparative cryo-EM of polymorphic channels and/or molecular dynamics simulations would be required (S7C Fig).

In addition to the GOF variant TPC2$^{M484L/P564L}$, endo-lysosomal patch-clamp experimentation revealed two further variants that showed moderately increased activation levels after stimulation with PI(3,5)P$_2$ compared to control: TPC2$^{K376R/L564P}$ and TPC2$^{G387D/L564P}$ (Fig 5C, 5D and 5J). Both combinations occur in several populations with similar frequency, but are the least common in African, and Native American populations (Figs 2 and 3). In contrast, the variant TPC2$^{V219I/L564P}$, mainly found in European populations, showed no significant difference compared to TPC2$^{L564P}$, nor did the V219I polymorphism appear to affect the GOF effect of its associated SNP M484L (Fig 5E, 5F and 5J). As noted previously, the SNP S681L was only found on the L564 background in contrast to all other homozygously occurring variations. TPC2$^{P564L/S681L}$ is predominantly found in East Asian populations. Albeit showing a trend towards increased basal activity, the activity level of the TPC2$^{P564L/S681L}$ variant upon stimulation with PI(3,5)P$_2$ was not significantly different from control, i.e., TPC2$^{P564L}$ (Fig 5G and 5J). By contrast, the blond-hair associated GOF variant TPC2$^{M484L/L564P}$ is barely found among East Asians and mainly present in European, Caucasian and West Asian populations (S5 Fig). All variants showed similar subcellular localization, highly correlating with lysotracker localization. In addition, no significant expression differences in western blot analysis were found (S6 Fig).

## Functional analysis of TPC2 SNPs in endogenously expressing human donor fibroblast samples

As a next step we wanted to confirm these findings in endogenously expressing cells. 136 human donor samples were genotyped [21] out of which, nine donor fibroblast samples were collected for subculture and further experimentation. All fibroblast donors were of Caucasian/European origin. In accordance with the interrogated genome databases, all donor samples

were homozygous for L564P (S8A Fig). Likewise, as expected, neither of the donors carried the East Asian variant S681L. Four donors (donor 1, 2, 5, and 8) were homozygous for M484L. All four were simultaneously either homo- or heterozygous for V219I, a SNP not significantly affecting basal or PI(3,5)P$_2$-stimulated channel activity. Of the four M484L donors, three were blond, and one brown-haired, in accordance with previously published data [21]. Another blond-haired donor reported previously [21] was homozygous for G734E (donor 6). Donors 3 and 9 were both heterozygous for K376R and G387D. Donor 4 was homozygous for K376R and donor 7 homozygous for G387D (S8A Fig). Endo-lysosomal patch-clamp experiments revealed significantly higher PI(3,5)P$_2$-stimulated TPC2 activity for all four M484L fibroblast samples compared to donor 3, which does not encode the M484L variation, nor any of the other variations in a homozygous manner (S8B and S8C Fig). Donor 4 (homozygous for K376R) and donor 7 (homozygous for G387D) both showed significantly higher PI(3,5)P$_2$-stimulated activity compared to control, but lower activity compared to the M484L samples. This mild GOF activity of both K376R and G387D is in accordance with overexpression data described above (Fig 5). In sum, these data reproduce key findings obtained from overexpressing HEK293 cells. The data further show complex TPC2 variation patterns in different individuals with several SNPs occurring at the same time. As expected, the combination of M484L and P564L was not found among the 136 samples, as it does not seem to occur naturally.

## Analysis of GWAS data

Publicly available GWAS databases were assessed and their inclusion of TPC2 polymorphisms interrogated [32,33]. Analysis of the NHGRI-EBI GWAS Catalog revealed associations of V219I, K376R, M484L, and G734E variants with hair color, which is in line with previous studies on the matter (see also S1 Table) [21,34]. Our meta-analysis revealed additional associations beyond hair pigmentation, showing V219I and M484L to be associated with a decreased risk for type 2 diabetes mellitus (T2DM). Furthermore, the G734E polymorphism appeared to be associated with bone mineral density. We investigated these findings further by interrogating the Type 2 Diabetes Knowledge Portal, a publicly available web-based GWAS tool providing access to human genetic information linked to T2DM and various other traits [33]. The top 5 significant associations of 7 TPC2 SNPs with phenotypes were collected (S2 Table) and are depicted in Fig 6A–6G. The data presented were collected from different consortia, such as the UK biobank eBMD, GIANT, GoT2D, and DIAMANTE, with sample sizes ranging from 1997 to 898130. V219I, M484L, and G734E appeared most strongly associated with hair pigmentation. V219I and M484L appeared protective against T2DM, and G734E was associated with increased bone mineral density and decreased height. Of interest, we found K376R to be associated with decreased bone mineral density and increased height, contrasting the associations of G734E. It should be further noted that the associations of V219I could be confused with associations due to the M484L polymorphism, as the two polymorphisms appear to be linked (Fig 6H). This observation is also supported by the donor fibroblast genotypes (S8 Fig). Taken together, TPC2 polymorphisms appear to be associated with various phenotypes, most consistently with hair color, T2DM, and bone mineral density.

## Discussion

We provide here the first in-depth analysis of variations of the endo-lysosomal cation channels TPC1, TPC2, TRPML1, TRPML2, and TRPML3, across different human populations including ancient human genomes. We analyzed genomes from three different genome datasets: The Simons genome data set, the 1000 genome project dataset, and the gnomAD dataset. All three datasets revealed higher occurrence of variations in TPC2 compared to the other endo-

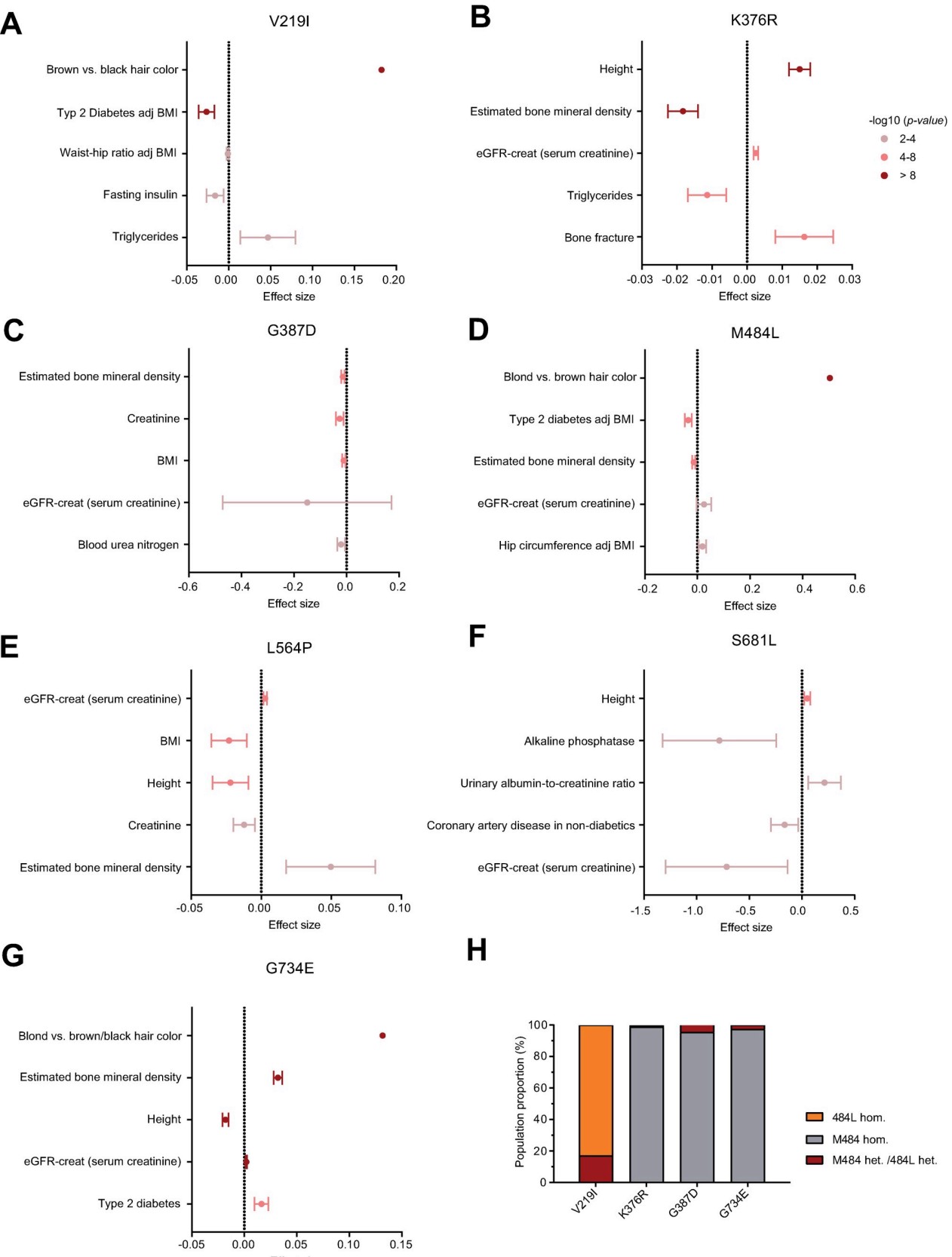

**Fig 6. GWAS analysis to assess TPC2 SNP disease association.** (A-G) The Type 2 Diabetes Knowledge Portal was employed to assess phenotypes associated with high-frequency TPC2 SNPs [33]. Odds ratios were converted into effect sizes to render the data format uniform, and standard error of the mean estimated from P values, sample sizes, and effect sizes. Forest plots were subsequently plotted for the top five associated traits, centered around an effect size of 0. Decreasing effect sizes would suggest inverse associations between the trait and the SNP (such as for M484L and type 2 diabetes), while increasing effect sizes suggest direct associations (such as for G734E and bone mineral density). Colors indicate statistical significance, with dark red data-points indicating genome-wide significance (P < $10^{-8}$). (H) We further investigated the association between V219I and other SNPs. We found V219I and M484L SNPs to be linked, with less linkage between M484L and other SNPs. Since M484L affects channel function more dramatically than V219I, we assume shared phenotypes associated with V219I to be largely attributable to the linked M484L.

lysosomal, non-selective cation channels TPC1, TRPML1, TRPML2, and TRPML3. Surprisingly, the TPC2 variant L564P occurs with such high frequency that the variant described hitherto as "wild-type" seems to be a rather rare variant. In addition, most other variations were found to occur on the 564P background rather than the L564 background, apart from S681L. Finally, all publicly available primate reference genomes carry proline instead of leucine at position 564. Functionally, by using endo-lysosomal patch-clamp experimentation we found that 564P has a surprising effect on the recently described M484L GOF variant of TPC2: without 564P, the M484L variation shows no GOF effect. Therefore, 564P appears to be a prerequisite for the M484L GOF effect, which is associated with a higher likelihood of light pigmentation and blond hair color, as previously reported [21,34]. The data further suggest that M484L has developed on the background of 564P, as it does not occur in combination with L564 in any of the analyzed genome samples. Furthermore, additional GOF variants of TPC2 were identified.

Given the many physiological and pathophysiological roles that TPC2 plays as outlined above, our data suggest that individuals with different TPC2 variations will likely show different susceptibilities to various diseases. Indeed, certain TPC2 polymorphisms appear to be correlated with bone mineral density (G734E, loss-of-inhibition) [35] and inversely correlate with type 2 diabetes mellitus (M484L, gain-of-function) [36], conditions previously associated with TPC2 function [23–25,37]. Beyond its association with diseases, the functional spectra of TPC2 also appear of interest pharmacologically. TPC2 was recently shown to be druggable, with two groups independently developing channel agonists either by drug repurposing [38] or drug screening for high-efficacy agonists [39]. Genetically informed drug design has been estimated to double the success rate of clinical drug development [40], for example the development of SLC30A8 antagonists as a potential therapy for type 2 diabetes [41,42]. Given the relevance of TPC2 in health and disease alongside its potential translational value, the presented findings suggest the channel to be a likely contributor to multigenic traits, with implications in basic research and pharmacogenomics. Additionally, the functional variability of TPC2 appears intriguing, and warrants further investigation into how individual TPC2 polymorphisms historically have been selected for by various selection pressures.

## Materials and methods

### Analysis of human genome variation data sets

The Simons Genome Project (SGDP) was accessed through the Cancer Genomics Cloud, and visualized using the Integrative Genomics Viewer through loading.BAM and.BAI URLs. Available metadata was recorded, identifying sample gender, ancestry, and continental affiliation. 1000 genome dataset and primate genomes were accessed through the ncbi.nlm.nih.gov website, the gnomAD dataset through gnomad.broadinstitute.org. All missense, nonsense or stop gained SNPs of each channel were included in the analysis. Due to their indirect affiliation with the channels, intronic variants were excluded from the analysis. Wild-type samples were assigned with the number 0, heterozygous samples with 0.5 und samples with homozygous

polymorphisms with the number 1, and mean allele frequencies (MAF) calculated. Published genomes of ancient samples were accessed as described in the respective publications [43–62]. Only genomes that had sequencing coverage of 5 or more for the amino acid 564 of TPCN2 were included.

## Human donor gDNA purification and sequencing

Genomic DNA was isolated from previously described, cultured primary human fibroblasts [21]. gDNA extraction was performed using the PureLink Genomic DNA Kit (Invitrogen) according to manufacturer's instructions. Loci of interest were amplified by PCR using Q5 polymerase (NEB) according to manufacturer's instructions, and primers suitable for polymorphic site amplification. Primers for amplifying and sequencing M484L and G734E loci were previously described [21]. The following primers for amplifying and sequencing the newly described TPC2 SNPs from gDNA were used: GCCCCCAGGGTTTCATTTGT (V219I forward), GAGTCCGCACATGGGTTAGG (V219I reverse, fragment size 427 bp), GGTG AAGTCAGTTTGCGCC (K376R forward), AAGGTGAGCGCCCTCGAA (K376R reverse, fragment size 444 bp), GCCATGATGGAGGTACCCG (G387D forward), GTTCCCCAGG TGATCAAGGG (G387D reverse, fragment size 381 bp), GCCTGACAGGCTGTGTGG (L564P forward), ACTCCAGTGCATAACCCGCC (L564P reverse, fragment size 303 bp), GTGCTCTTGCTTTGCTCATC (S681L forward), CGACTCTCCCATCAAAGTTCC (S681L reverse, fragment size 246 bp). Amplified PCR products were purified by agarose gel electrophoresis, band excision, and gel extraction using the QIAquick Gel Extraction Kit (Qiagen) according to manufacturer's instructions. Subsequently, samples were sequenced using forward and reverse amplification primers to identify TPC2 genotypes.

## Endo-lysosomal patch-clamp experiments

For whole-endolysosome manual patch-clamp recordings, cells were treated with vacuolin (1 μM) overnight [63]. Currents were recorded using an EPC-10 patch-clamp amplifier (HEKA, Lambrecht, Germany) and PatchMaster acquisition software (HEKA). Data were digitized at 40 kHz and filtered at 2.8 kHz. Fast and slow capacitive transients were cancelled by the compensation circuit of the EPC-10 amplifier. Recording glass pipettes were polished and had a resistance of 4–8 MΩ. Cytoplasmic solution contained 140 mM K-MSA, 5 mM KOH, 4 mM NaCl, 0.39 mM $CaCl_2$, 1 mM EGTA and 10 mM HEPES (pH was adjusted with KOH to 7.2). Luminal solution contained 140 mM Na-MSA, 5 mM K-MSA, 2 mM Ca-MSA, 1 mM $CaCl_2$, 10 mM HEPES and 10 mM MES (pH was adjusted with NaOH to 4.6). In all experiments, 500-ms voltage ramps from -100 to +100 mV were applied every 5 s, holding potential at -60 mV. The current amplitudes at -100 mV were extracted from individual ramp current recordings. All statistical analyses were performed using Origin8 software.

## Cell culture and mutagenesis

HEK293 cells were maintained in DMEM supplemented with 10% FBS, 100 U penicillin/mL, and 100 μg streptomycin/mL at 37˚C and 5% $CO_2$. For patch clamp experiments, cells were plated on poly-L-lysine (0,1%)-coated glass coverslips 60–96 hours before experimentation, and transiently transfected with Turbofect (ThermoFisher) according to the manufacturer's protocols and used for patch clamp experiments 36 hours after transfection.

Human genomic DNA sampling and human fibroblast isolation were approved by the Ludwig-Maximilians-Universität Ethics Committee (headed by Prof. Dr. Eisenmenger; reference no. 254–16). Acquisition of human material was performed after obtaining written informed consent by the donors as described previously [21]. Primary fibroblasts were isolated by Prof.

Dr. Carola Berking and colleagues (Department of Dermatology, Ludwig-Maximilians-Universität München) from the skin of healthy adult donors. Epidermis was separated from dermis using dispase II (10 mg/mL in PBS, pH 7.2–7.4, D4693; Sigma) and dermis was digested in collagenase (1 mg/mL in DMEM, C0130; Sigma) for 22 h at room temperature. Fibroblasts were cultured in DMEM with glutamine (Life Technologies, Inc.) and 10% FBS (FBS Superior, S0615; Biochrom). Human TPC2 SNPs were generated by site-directed mutagenesis as previously described [21]. The following oligonucleotide primers were used for site-directed mutagenesis PCRs to generate newly identified TPC2 SNP variants or other mutants: V219I forward: CGGAAATGGCCAGCATCGGGCTGCTGCTGGCC; V219I reverse: GGCCAG CAGC AGCCCGATGCTGGCCATTTCCG; K376R forward: GCTGGACAGCTCCCACAG ACAGGCCATGATGGAG; K376R reverse: CTCCATCATG GCCTGTCTGTGGGAGCTGT CCAGC; G387D forward: AAGGTGCGTTCCTACGACAGTGTTCTGCTGTCAGC; G387D reverse: GCTGACAGCA GAACACTGTCGTAGGAACGCACCTT; P564L forward: GCGT ATCATCCCCAGCATGAAGCTGATGGCCGTGGTGGCC; P564L reverse: GGCCACCA CGGCCATCAGCTTCATGCTGGGGATGATACGC; S681L forward: GTGGTGGCTGGTG TTGTCTGTCATCTGGG; S681L reverse: CCCAGATGACAGACAACACCAGCCACCAC. L564A forward: ATCATCCCCAGCATGAAGGCGATGGCCGTGGTG; L564A reverse: CACCACGGCCATCGCCTTCATGCTGGGGATGAT; L564G forward: ATCATCCCCAG CATGAAGGGGATGGCCGTGGTG; L564G reverse: CACCACGGCCATCCCCTTCATG CTGGGGATGAT; M562P forward: GCGTATCATCCCCAGCCCGAAGCTGATGGCCG TG; M562P reverse: CACGGCCATCAGCTTCGGGCTGGGGATGATACGC; K563P forward: TATCATCCCCAGCATGCCGCTGATGGCCGTGGTG; K563P reverse: CACCACG GCCATCAGCGGCATGCTGGGGATGATA; M565P forward: CCAGCATGAAGCTGCCG GCCGTGGTGGCCA; M565P reverse: TGGCCACCACGGCCGGCAGCTTCATGCTGG.

## TPC2 genotype-phenotype linkage analysis

The high-frequency TPC2 polymorphisms were investigated for associations with phenotypes using the following identifiers: rs72928978 (V219I), rs3750965 (K376R), rs61746574 (G387D), rs35264875 (M484L), rs2376558 (L564P), rs78034812 (S681L), and rs3829241 (G734E). GWAS results obtained from the NHGRI-EBI GWAS Catalog were compiled in S1 Table for polymorphisms showing phenotype associations (V219I, K376R, M484L, and G734E) [32]. TPC2 genotypes were next investigated in the Type 2 Diabetes Knowledge Portal, which includes numerous phenotypes beyond type 2 diabetes mellitus (T2DM) [33]. Where necessary, odds ratios were converted into effect sizes as described previously [64]. For plotting forest plots, SEM values were extracted using R version 1.2 as follows: Traits, effect sizes, sample sizes, were extracted, reformatted, and stored in R dataframes for statistical analysis. T-statistics were calculated using the qt() function, providing reported P values and degrees of freedom as inputs. SEM was subsequently calculated upon dividing effect sizes with the respective T-statistic. Forest plots of the top five associations for each polymorphism were plotted using GraphPad Prism v8, and data-points colored according to P values reported by the Type 2 Diabetes Knowledge Portal.

## Western blotting

For Western blot experiments transiently transfected HEK293 cells were washed twice with 1x PBS and pellets were collected. Total cell lysates were obtained by solubilizing in TRIS HCl 10 mM pH 8.0 and 0.2% SDS supplemented with protease and phosphatase inhibitors (Sigma). Protein concentrations were quantified via Bradford assay. Proteins were separated via a 7% sodium dodecyl sulphate polyacrylamide gel electrophoresis (SDS-PAGE; BioRad) and

transferred to polyvinylidene difluoride (PVDF; BioRad) membranes. Membranes were blocked with 5% bovine serum albumin (Sigma) diluted in Tris Buffered Saline supplemented with 0.5% Tween-20 (TBS-T) for 1 h at room temperature (RT), then incubated with primary antibody at 4˚C overnight. Then, membranes were washed with TBS-T and incubated with horseradish peroxidase (HRP) conjugated anti-mouse or anti-rabbit secondary antibody at RT for 1 h. Membranes were then washed and developed by incubation with Immobilon Crescendo Western HRP substrate (Merck) and by using an Odyssey imaging system (LI-COR Biosciences). Quantification was carried out using unsaturated images on ImageJ 1.52a software. The following primary antibodies were used: Rabbit polyclonal Anti-GFP, abcam, Cat# ab6556 (1:5000 in TBS-T with 5% BSA), mouse monoclonal Anti-β-tubulin, Cell Signaling, Cat# 86298S (1:1000 in TBS-T with 5% BSA). The following secondary antibodies were used: Anti-rabbit IgG, HRP-linked, Cell Signaling, Cat# 7074S (1:5000 in TBS-T with 5% BSA) and anti-mouse IgG, HRP-linked, Cell Signaling, Cat# 7076S (1:5000 in TBS-T with 5% BSA).

### Statistical analysis

Details of statistical analyses and n values are provided in the Materials and Methods or the figures or figure legends. Statistical analyses were carried out using Origin 8 and GraphPad Prism 8. All error bars are depicted as mean ± SEM. Statistical significance is denoted on figures as outlined in the legends. Statistics carried out on datasets upon obtaining three independent biological and technical replicates.

### Supporting information

**S1 Fig. Mean allele frequency of SNPs in other endo-lysosomal cation channels in the SGDP data set.** (A-D) Shown are the mean allele frequencies of all SNPs that were found in TRPML1-3 or TPC1 upon investigating the Simons Genome Diversity Project data set [26]. (EPS)

**S2 Fig. Mean allele frequency of SNPs in TPC2 and other endo-lysosomal cation channels in the 1000GP data.** (A-E) Shown are the mean allele frequencies of all SNPs that were found in TRPML1-3 or TPC1 and 2 when investigating the 1000 genomes project [27]. Homozygously occurring SNPs marked in red. (F) Percentage of polymorphisms for TRPML1-3 and TPC1 and 2, relative to the coding sequence length for each gene (left: all; right: only homozygous SNPs)
(EPS)

**S3 Fig. Mean allele frequency of SNPs in TPC2 and other endo-lysosomal cation channels in the gnomAD data set.** (A-E) Shown are the mean allele frequencies of all homozygous SNPs that were found in TRPML1-3 and TPC1 and 2 when investigating the gnomAD data set [28]. (F) Mean allele frequency of each SNP grouped by continent.
(EPS)

**S4 Fig. Analysis of ancient genome and primate samples.** (A) The presented map was generated using data obtained from published ancient genome sequences [43–62]. Sequencing results of 39 fossils of modern humans with a minimum coverage of 5 were analyzed. Square indicates where the fossil was found. Filled square represents a homozygous 564P sample, empty square represents a homozygous L564 sample, and semi-filled square represents a heterozygous sample. The color scheme encodes the estimated age of each fossil. (B) Shown are the 39 samples from ancient humans (blue) and an additional three samples from homo neanderthalensis (green) and one denisovan sample (red) on a time scale. Homozygous L564 samples are shown as empty triangles, homozygous 564P samples as filled triangles, and

heterozygous L564/564P as semi-filled triangles (C) Comparison of the TPC2 protein sequence in the human reference genome with other primate reference genomes. The neighbor-joining tree was created with the software MEGA X based on the FASTA sequences from the NCBI Gene database [65].
(EPS)

**S5 Fig. Maps of the worldwide distribution of TPC2$^{M484L}$ in the 1000GP and the gnomAD data set.** (A) Each circle represents one of the 26 populations of the 1000GP data set (indigenous or non-indigenous (other descent) as indicated) [27,28]. White color represents the fraction of homozygous carriers for TPC2$^{M484}$, red color represents the fraction homozygous for TPC2$^{484L}$, and pink color represents the heterozygous population. (B) Each circle represents a population as indicated on the gnomAD website. Color-coding as in A.
(EPS)

**S6 Fig. Analysis of expression and subcellular localization of TPC2 variants.** (A) Representative confocal microscopy images (Zeiss LSM 880 Airyscan) of different TPC2 variants transiently transfected in HeLa cells. Scale bar = 10 μm. (B) Colocalization of human TPC2 variants with lysotracker deep red (DR), quantified using Manders' coefficient or Pearson correlation coefficient (PCC). (C, D) Western blot analysis of the different TPC2 variants transiently transfected in HEK293 cells. Shown are (C) average relative expression levels (normalized to β-tubulin, n = 6 independent experiments, mean ± SEM) and (D) one representative blot. One-way ANOVA followed by Tukey's post-hoc test was applied to test for statistical significance
(EPS)

**S7 Fig. Analysis of TPC2 mutations in the IIS4-S5 linker region.** (A) Cartoon showing the location of the mutated amino acids in the IIS4-S5 linker region and mutation M484L. (B) Representative PI(3,5)P$_2$ (1 μM) activated current densities in vacuolin-enlarged lysosomal vesicles isolated from HEK293 cells expressing TPC2 mutants K563P or M565P in combination with M484L and P564L, each. (C) Model based on the recently resolved human TPC2 (M484/564P) cryo-EM structure [31]. The IIS4-S5 linker has been implicated in channel gating, as an extension of the IIS4-S5 linker appears necessary to provide space for pore dilation and channel opening. Substituting the helix-initiating 564P with a leucine would dramatically affect this linker helix extension. Our model proposes that M484L is amplifying the effect of PI(3,5)P$_2$ activation, requiring signal transduction through L564P to result in pore dilation.
(EPS)

**S8 Fig. TPC2 SNPs in human donor fibroblast samples.** (A) Genotyping results of human donor fibroblast samples. (B) Representative PI(3,5)P$_2$ (10 μM) activated current densities in vacuolin-enlarged lysosomal vesicles isolated from different donor fibroblasts. (C) Statistical summary of lysosomal patch-clamp data from human donor samples. Shown are average current densities (mean ± SEM) at -100 mV. Unpaired t-tests were applied. $^*$p < 0.05 and $^{**}$p < 0.01
(EPS)

**S1 Table. TPC2 polymorphisms reaching genome-wide significance in GWAS studies.** In the GWAS catalog four TPC2 polymorphism are listed that reach significant association in genome-wide association studies. The "OR" refers to the allelic odds ratio in each study. Further details can be found in the respective publications.
(XLSX)

**S2 Table. Data obtained from the GWAS Catalog and Type 2 Diabetes Knowledge Portal.**
For each homozygous occurring TPC2 polymorphism, traits were sorted by statistical significance (p value), and the top five traits were analyzed further. Effect sizes were used as reported or calculated from the odds ratio. Standard error of the mean was calculated using the qt() function in R. The data was subsequently used to generate genotype/phenotype-association forest plots, as illustrated in Fig 6.
(XLSX)

## Author Contributions

**Conceptualization:** Christian Grimm.

**Data curation:** Julia Böck, Einar Krogsaeter, Yu-Kai Chao.

**Formal analysis:** Julia Böck, Einar Krogsaeter, Marcel Passon, Yu-Kai Chao.

**Funding acquisition:** Christian Grimm.

**Methodology:** Julia Böck, Einar Krogsaeter, Marcel Passon, Yu-Kai Chao, Sapna Sharma, Harald Grallert, Annette Peters.

**Resources:** Annette Peters, Christian Grimm.

**Supervision:** Christian Grimm.

**Validation:** Julia Böck, Einar Krogsaeter, Marcel Passon, Christian Grimm.

**Visualization:** Julia Böck, Einar Krogsaeter, Marcel Passon, Christian Grimm.

**Writing – original draft:** Christian Grimm.

**Writing – review & editing:** Sapna Sharma, Christian Grimm.

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
