## [Decision Letter · Decision Letter 0]

3 Jul 2020

Dear Dr Grimm,

Thank you very much for submitting your Research Article entitled 'Human genome diversity data reveal that L564P is the predominant TPC2 isoform and a prerequisite for the blond hair associated M484L gain-of-function effect.' to PLOS Genetics. Your manuscript was fully evaluated at the editorial level and by independent peer reviewers. The reviewers appreciated the attention to an important problem, but raised some substantial concerns about the current manuscript. Based on the reviews, we will not be able to accept this version of the manuscript, but we would be willing to review again a much-revised version. We cannot, of course, promise publication at that time.

If you decide to revise the manuscript for further consideration at PLOS Genetics, please aim to resubmit within the next 60 days, unless it will take extra time to address the concerns of the reviewers, in which case we would appreciate an expected resubmission date by email to plosgenetics@plos.org.

[LINK]

We are sorry that we cannot be more positive about your manuscript at this stage. Please do not hesitate to contact us if you have any concerns or questions.

Yours sincerely,

Elena Oancea

Guest Editor

PLOS Genetics

Gregory Barsh

Editor-in-Chief

PLOS Genetics

Reviewer's Responses to Questions

**Comments to the Authors:**

**Reviewer #1**: In this manuscript, Böck et al. analyzed the available human genetic sequences and characterized several polymorphisms in the two-pore channel TPCN2 gene. The authors have previously reported that one such SNP (p.M484L) associated with hair color is a gain-of-function variation, with channel harboring the 484L allele generating larger ionic currents than the one harboring 484M. In this paper, the authors report that the channel current enhancement mediated by p.M484L is influenced by another locus at residue 564. TPC2 carrying the major allele at this locus (564P), as previously reported, can be potentiated by the p.M484L variation, but TPC2 carrying the minor allele at this locus (564L) can’t. The authors further analyzed the GWAS data and discovered potential linkage of several diseases to variations in this gene.

The data is interesting that the p.M484L gain-of-function variation requires the presence of the major allele at position 564; the analysis is of potential use to link TPC2 variations to some of the diseases. The results however appear to be preliminary and need to be more convincing.

(1) The p.M484L variation has been previously linked to hair color. The authors conclude in this paper that in the 564L background, the p.M484L variation has little effect on the TPCN2 function, suggesting that the p.M484L in the 564L background will not be linked to hair color. The authors should examine the data and see whether such a link indeed does not exists in the 564L background.

(2) There is generally a lack of mechanism explaining the apparent influence of residue 564 on the potentiation of channel current by p.M484L. Is it because of a change in channel expression level, or in the sensitivity in the activation by PI(3,5)P2 or the inhibition by ATP?

(3) Most of the functional analysis is from overexpression. There is limited data from the analysis of SNP-associated endogenous TPC function from cultured fibroblasts, and it is not that convincing due to the limited number of recordings (n=3-6). In addition, the 9 donors analyzed in Fig. S6 all harbor 564P and therefore can’t be used to test the effect of 564L. In order to test the effect of 564L on endogenous TPC2 function, the authors should analyze more fibroblasts and identify individuals carrying the minor allele. Given the fairly high frequency of the minor allele (~13%), this might be doable. Alternatively, even better, the authors can perhaps knock-in the 564L minor allele into one of the existing fibroblast cell line and analyze the effect.

**Reviewer #2**: This study explores the genetic and functional analysis of polymorphisms found in the endo-lysosomal cation channel TPC2. They used publically available dataset of human genome sequences to determine the polymorphism profiles and frequencies across the populations represented in the datasets. They then used endo-lysosomal patch-clamp analyses to show that the GOF M484L Variant only acts as a GOF variant on the 564P but not on the 564L background. The also compare activity in donor fibroblasts samples and find suggest that there are TPC2 variation patterns across the globe that alter TPC2 to become more or less active with and/or without stimulation. This is an important characterization of a gene with significant medical and global importance.

Comments:

Page 9. The authors state that there is a higher variation frequency for TPC2 than for similar channels. While it is true that there more polymorphisms found for TPC2 than for other channels I am not sure if this is significant. Did they control for exon size and look at noncoding/intronic region variation of these genes to make this statement. If not maybe just state the frequencies without a qualifier as it being higher variation. Maybe say the most variation was seen in.

In transfection and endogenous patch clamp studies, how did authors control for equal protein levels and correction cellular locations of TCP2 variant?

PAGE 11. The authors stat that several fibroblasts were assessed. Please insert the actual number.

**Have all data underlying the figures and results presented in the manuscript been provided?**

Reviewer #1: Yes

Reviewer #2: Yes

PLOS authors have the option to publish the peer review history of their article (what does this mean?). If published, this will include your full peer review and any attached files.

Reviewer #1: No

Reviewer #2: No

---

## [Decision Letter · Decision Letter 1]

29 Oct 2020

Dear Dr Grimm,

We are pleased to inform you that your manuscript entitled "Human genome diversity data reveal that L564P is the predominant TPC2 isoform and a prerequisite for the blond hair associated M484L gain-of-function effect." has been editorially accepted for publication in PLOS Genetics. Congratulations!

Yours sincerely,

Elena Oancea

Guest Editor

PLOS Genetics

Gregory Barsh

Editor-in-Chief

PLOS Genetics

Reviewer's Responses to Questions

**Comments to the Authors:**

Reviewer #1: The authors have satisfactorily addressed my previous comments.

Reviewer #2: the authors have address my concerns.

**Have all data underlying the figures and results presented in the manuscript been provided?**

Reviewer #1: Yes

Reviewer #2: Yes

PLOS authors have the option to publish the peer review history of their article (what does this mean?). If published, this will include your full peer review and any attached files.

Reviewer #1: No

Reviewer #2: No

**Data Deposition**

http://datadryad.org/submit?journalID=pgenetics&manu=PGENETICS-D-20-00676R1

**Press Queries**

---

## [Editor Report · Acceptance letter]

14 Jan 2021

PGENETICS-D-20-00676R1 

Human genome diversity data reveal that L564P is the predominant TPC2 isoform and a prerequisite for the blond hair associated M484L gain-of-function effect. 

Dear Dr Grimm, 

We are pleased to inform you that your manuscript entitled "Human genome diversity data reveal that L564P is the predominant TPC2 isoform and a prerequisite for the blond hair associated M484L gain-of-function effect." has been formally accepted for publication in PLOS Genetics! Your manuscript is now with our production department and you will be notified of the publication date in due course.

With kind regards,

Melanie Wincott

PLOS Genetics

On behalf of:
